# Controlled Construction of Stable Network Structure Composed of Honeycomb-Shaped Microhydrogels

**DOI:** 10.3390/life8040038

**Published:** 2018-09-20

**Authors:** Masayuki Hayakawa, Satoshi Umeyama, Ken H. Nagai, Hiroaki Onoe, Masahiro Takinoue

**Affiliations:** 1RIKEN Center for Biosystems Dynamics Research, Kobe, Hyogo 650-0047, Japan; masayuki.hayakawa@riken.jp; 2Department of Computer Science, School of Computing, Tokyo Institute of Technology, Yokohama, Kanagawa 226-8502, Japan; hozzyyyyyyy@gmail.com; 3School of Materials Science, Japan Advanced Institute of Science and Technology, Nomi, Ishikawa 923-1292, Japan; k-nagai@jaist.ac.jp; 4Department of Mechanical Engineering, Keio University, Yokohama, Kanagawa 223-8522, Japan; onoe@mech.keio.ac.jp

**Keywords:** cell communication, artificial cell/tissue, microhydrogel network, microfluidics

## Abstract

Recently, the construction of models for multicellular systems such as tissues has been attracting great interest. These model systems are expected to reproduce a cell communication network and provide insight into complicated functions in living systems./Such network structures have mainly been modelled using a droplet and a vesicle. However, in the droplet and vesicle network, there are difficulties attributed to structural instabilities due to external stimuli and perturbations. Thus, the fabrication of a network composed of a stable component such as hydrogel is desired. In this article, the construction of a stable network composed of honeycomb-shaped microhydrogels is described. We produced the microhydrogel network using a centrifugal microfluidic technique and a photosensitive polymer. In the network, densely packed honeycomb-shaped microhydrogels were observed. Additionally, we successfully controlled the degree of packing of microhydrogels in the network by changing the centrifugal force. We believe that our stable network will contribute to the study of cell communication in multicellular systems.

## 1. Introduction

In a living multicellular system, such as tissue, sophisticated functions are realized via the organization of individual cells. Recently, the experimental construction of models for living multicellular systems has attracted considerable interest. The use of such model systems allows the extraction of the essence of highly orchestrated but complicated functions in living systems, and helps to understand them. In particular, recent development of a network model system that is useful for understanding cell communications has been remarkable. Cell communication is essential for many important and vital functions in biological systems. For example, gap junctions perform the exchange of small molecules and ions, which is vital for homeostasis maintenance [1,2]; cell communication through molecular diffusion is an important factor in the pattern formation of skin [3,4]. Using diffusively-connected networks of droplets, various pattern formations [5,6], chemical communications via quorum signalling molecules [7], and a differentiation process [8] have been investigated. In the surfactant bilayer network [9] and networks of aqueous droplets [10,11], the construction of pore-connected pathways conducting an electric current was demonstrated. Furthermore, a three-dimensional (3D) network composed of lipid-coated aqueous droplets was realized using 3D-printing technology; the self-folding of droplet networks was demonstrated by an osmotic interaction between droplets [12]. Further, although the network size is not large, recently, the construction of vesicle networks has been in development [13,14]. Currently, droplet and vesicle networks are an important platform for understanding the communication in living multicellular systems. However, because the contact boundaries of the droplets or the vesicles are unstable, problems such as unexpected coalescence due to external factors, e.g., electric current, and the difficulty of handling the network, cannot be neglected. Challenging studies reported a network that was embedded in an organogel block to stabilize the whole network structure [15], but the problem of instability at the boundary between droplets remained. Another group proposed highly-stable communicating droplets encapsulated in a hydrogel [16]. However, it is difficult to construct a network structure as large as a living multicellular system, because the size of the hydrogel encapsulating the droplet is in the millimeter scale. Hence, previous studies have not succeeded in the construction of a network model for multicellular systems with stability against external stimuli. To achieve this, construction of a micrometer-scale network model composed of a stable component (e.g., hydrogels) is required. Here, we propose a fabrication method for a network structure composed of stable honeycomb-shaped microhydrogels. Our method can control the size and packing degree of each honeycomb-shaped microhydrogel. In this paper, we first describe the fabrication of the honeycomb microhydrogel network with the control of the size and the packing degree. In addition, molecular diffusion among the honeycomb-shaped microhydrogels was investigated. Finally, the fabrication of a triple-layered honeycomb microhydrogel network and a network composed of double-faced honeycomb microhydrogels was demonstrated. The method described in this paper will be useful for constructing a sophisticated experimental model and understanding the essence of cell communication in living systems.

## 2. Materials and Methods

### 2.1. Materials and Experimental Setup

We constructed the honeycomb microhydrogel network using a centrifuge-based microfluidic device [17,18] and a photosensitive polymer solution. The microfluidic device consisted of a theta-shaped glass capillary (TST150-6, World Precision Instruments, Sarasota, FL, USA), a capillary holder, and a sampling microtube (1.5 mL microtube CF-0150, BIO-BIK, Osaka, Japan) (Figure 1a). The glass capillary and the capillary holder were processed according to a previously reported study [17,18]. In this study, the orifice diameter of a tip of the theta-shaped glass capillary was tuned to 30 μm (Figure 1b). The tuned theta-shaped glass capillary was used in all experiments to standardize experimental conditions. The inside of the glass capillary was filled with a photosensitive polymer solution comprising 34% (*v*/*v*) polyethylene(glycol) diacrylate (PEGDA) (Mn = 700) (ALD455008, Sigma–Aldrich, St. Louis, MO, USA), 6% (*v*/*v*) 2-hydroxy-2-methylpropiophenone (Darocur) (ALD405655, Sigma–Aldrich), and 2.4% (*v*/*v*) glycerol (075-00616, Wako Pure Chemical Industries, Osaka, Japan). In addition, depending on the experiment, a fluorescein sodium salt (F6377, Sigma–Aldrich), a dispersion of red-fluorescent microbeads (Fluoresbrite Carboxylate Microspheres 0.10 μm NYO, Polyscience, Inc., Warminster, PA, USA), and green-fluorescent microbeads (Fluoresbrite Carboxylate Microspheres, 0.10 μm YG, Polyscience, Inc.) were added to the photosensitive polymer solution. Although the fluorescent microbeads sometimes aggregated in the photosensitive polymer solution, the aggregates did not affect the formation of the honeycomb microhydrogel network. The sampling tube was filled with 170 μL of mineral oil (23334-85, Nacalai Tesque, Inc., Kyoto, Japan) containing a surfactant: 3% (*v*/*v*) ABIL EM 90 (Evonik Degussa Japan Co., Ltd., Osaka, Japan).

### 2.2. Fabrication and Analysis of Honeycomb Hydrogel Network

Figure 1c–e shows the fabrication of the honeycomb microhydrogel network. To enable fluorescence observations, the fluorescein sodium salt was added to the photosensitive polymer solution (final concentration: 1 mM), and they were introduced on both sides of the theta-shaped glass capillary. The assembled microfluidic device was placed in a swing rotor centrifuge (ATT 101, HITECH Co., Ltd., Tokyo, Japan) (Figure 1c) and subjected to a centrifugal force *F*_G_ (1183× *g*) for 1 min under ultraviolet (UV) light irradiation (λ = 365 nm) (lamp: HLR100T-2, power supply: HB100A-2, SEN LIGHTS Co., Ltd., Osaka, Japan) from the top of the centrifuge. Under *F*_G_, droplets of the photosensitive polymer solution were formed at the tip of the capillary, and were discharged toward the bottom of the sampling tube through air and oil phases (Figure 1d,e). To avoid the solidification of the photosensitive polymer solution in the glass capillary, all but the bottom of the sampling tube was wrapped by aluminium foil (Figure 1d,e).

The honeycomb microhydrogel network was removed from the sampling tube using tweezers, and observed using confocal laser scanning microscopy (CLSM) (FV 3000, Olympus Co., Tokyo, Japan). The area of the honeycomb-shaped microhydrogel *S*_gel_ and the gap area between them *S*_gap_ were manually measured using the software Image J (National Institutes of Health, New York, NY, USA, 2015, 1.50a). We defined a size parameter *d* to characterize the size of the honeycomb-shaped microhydrogel; *d* was calculated using *d* = 2(*S*_gel_/π)^1/2^. 

The triple-layered honeycomb microhydrogel network was constructed by stacking layers of honeycomb microhydrogel networks. After the fabrication of the first layer of the honeycomb microhydrogel network, the glass capillary was changed. Then, the second honeycomb microhydrogel network was fabricated on the first layer. Finally, the third layer was fabricated in the same way. To clearly confirm the layer boundaries, each honeycomb microhydrogel network was fabricated using the photosensitive polymer solution containing different fluorescent microbeads (first layer: 0.45% (*v*/*v*) green-fluorescent microbeads and 0.045% (*v*/*v*) red-fluorescent microbeads; second layer: 0.5% (*v*/*v*) red-fluorescent microbeads; third layer: 0.5% (*v*/*v*) green-fluorescent microbeads). Here, we blended the green- and red-fluorescent microbeads to generate a yellow fluorescence color for the first layer.

A double-faced honeycomb microhydrogel network was constructed using the glass capillary with different solutions in each compartment of the theta-shaped capillary. Here, both compartments of the capillary contained the photosensitive polymer solution with 1% (*v*/*v*) sodium alginate (196–13,325, Wako Pure Chemical Industries), and these solutions were colored by adding 0.25% (*v*/*v*) fluorescent microbeads (green and red). The Reynolds number decreased owing to an increase in the viscosity as a result of the addition of the sodium alginate. This prevented the photosensitive polymer solution in each compartment of the discharged droplet from mixing together.

### 2.3. Experiment and Analysis of Fluorescence Recovery after Photobleaching (FRAP) 

FRAP analysis was performed in the condition of a non-gel aqueous solution, a non-compartmentalized bulky hydrogel, and the honeycomb microhydrogel networks using a FRAP system integrated in the CLSM. Here, the fluorescein sodium salt was used as a fluorescence molecule. We examined the diffusion of the fluorescence molecule in the Milli-Q water (Merck KGaA, Darmstadt, Germany), the bulk hydrogel, and the honeycomb microhydrogel network. The Milli-Q water contained 10 mM fluorescein sodium salt. Both the hydrogels were fabricated using identical photosensitive polymer solution (as described in Section 2.1) containing 10 mM fluorescein sodium salt. The bulk hydrogel was prepared via 1 min of homogeneous UV irradiation to 7.5 μL of the photosensitive polymer solution in the sampling tube. The honeycomb microhydrogel network was fabricated under *F*_G_ = 1183× *g*. In the Milli-Q water and the bulk hydrogel, the fluorescence was circularly bleached with a radius of 50 μm, whereas in the honeycomb microhydrogel network, the fluorescence of the whole area of one of the honeycomb-shaped microhydrogels was bleached. After the photobleaching, fluorescence recovery was observed for 5 min, and the fluorescence intensity *I* at each time *t* was measured. The normalized fluorescence intensity at each time was defined as *I*_n_ (*t*) = *I*(*t*)/*I*(0). The FRAP experiment were performed three times in each condition, and the mean values of *I*_n_ (*t*) were calculated at each *t*.

## 3. Results and Discussion

### 3.1. Fabrication of Network Structure

Figure 2a shows the sampling tube immediately after the fabrication of the honeycomb microhydrogel network. At the bottom of the sampling tube, a layer composed of stacked microhydrogels was observed (orange rectangle in Figure 2a). Figure 2b,c show CLSM images of the honeycomb microhydrogel network; a densely packed network structure composed of honeycomb-shaped microhydrogels was observed.

It is speculated that this honeycomb microhydrogel network was formed by the balance between the time for the polymerization and the *F*_G_-induced deformation of the droplet at the bottom of the sampling tube. When *F*_G_ was applied, a sphere packing structure composed of the droplets was formed in a two-dimensional plane on the bottom of the sampling tube. Because it takes a certain time for polymerization, most of the droplets in the packing structure were incompletely polymerized. Since the elasticity of the moderately polymerized droplets was small, they underwent deformation by *F*_G_ and lateral contact with neighboring droplets as the density of the droplets in the packing structure increased. After a maximum density was achieved, another sphere packing structure was formed on the existing packing structure, and each droplet in the sphere packing structure deformed in the same manner. Eventually, the droplets were completely polymerized, resulting in the formation of the honeycomb microhydrogel network.

The shape of the microhydrogels outside the broken line was not a perfect honeycomb-shape (Figure 2b). This imperfection is considered to be due to the excessive deformation of droplet. When the droplet was introduced in the existing sphere packing structure, the droplet collided with the moderately polymerized droplets in the packing structure. Because the droplet was in the liquid state immediately after it was introduced into the structure, the droplet could undergo excessive deformation, and this could induce incomplete honeycomb-shape formation. Another possibility could be that the imperfect honeycomb-shape was imaged because the stacked microhydrogel did not exist in the same plane.

### 3.2. Control of Structure in Hydrogel Network by Changing Centrifugal Force

As shown in Figure 2, it is speculated that one of the factors of the honeycomb packing was *F*_G_-induced deformation. Hence, we demonstrated the control of the hydrogel network structure by changing *F*_G_. We fabricated the honeycomb hydrogel network under *F*_G_ = 67,312, and 1183× *g*, as shown in Figure 3a–c, respectively. From those CLSM images of the hydrogel network under each *F*_G_, the area *S*_gel_, the gap area *S*_gap_ and the size parameter *d* of the individual hydrogel were obtained. In the lower images in Figure 3a–c, *S*_gap_ are colored blue. Histograms of *d* in *F*_G_ = 67,312, and 1183× *g* are shown in Figure 3d, indicating that size distributions were narrow in each *F*_G_. We attributed this size controllability to our centrifuge-based microfluidic device [17,18]. As shown in Figure 3e, the mean of size parameter <*d*> decreased as *F*_G_ increased, indicating that the size of the hydrogels in the network structure was well controlled. In the condition of *F*_G_ = 1183× *g*, the droplet-discharging frequency was high. In this situation, more droplets can be additionally introduced in the packing layer, which increased the force of the lateral contact. As a result, <*d*> decreased.

On the other hand, when *F*_G_ = 67× *g*, the droplet-discharging frequency was low. In this case, it is speculated that most of the droplets polymerized completely before the formation of the sphere packing. Because the completely polymerized droplets cannot be deformed, additional droplets were not packed in the existing layer. As a result, <*d*> was larger than <*d*> in *F*_G_ = 1183× *g*. Furthermore, to analyze the relationship between *F*_G_ and the degree of packing, we measured the gap area *S*_gap_ (blue areas in Figure 3a–c), and *R* = *S*_gap_/*S*_gel_ was defined. As shown in Figure 3f, *R* decreased as *F*_G_ increased, indicating that *R* was also controlled by changing the applied *F*_G_. We attributed this controllability of the network structure to the stability of the hydrogel. As previously indicated, the network structure was controlled using the deformation of the individual microhydrogels. The hydrogel stably retained its shape as it solidified, and the packing degree of the constructed network was thus well controlled.

### 3.3. Inhibition of Molecular Diffusion in Honeycomb Hydrogel Network

We performed FRAP analysis to test whether our network structure can be applied to the model of the multicellular system, i.e., whether the contents can be confined within each individual honeycomb-shaped microhydrogel, as in living cells. Here, we show that the honeycomb-shaped microhydrogel in the network inhibited molecular diffusion. CLSM images of photobleaching at *t* = 0 s and recovery at *t* = 295 s in each medium are presented in Figure 4a–c (a: the non-gel aqueous solution, b: the non-compartmentalized bulky hydrogel, c: the honeycomb microhydrogel network). In each image, the photobleached area is enclosed by a red line. Figure 4d shows fluorescent recovery curves obtained from each experiment (green: the non-gel aqueous solution, red: the non-compartmentalized bulky hydrogel, blue: the honeycomb hydrogel network), and the error bars depicted in 11.1 s intervals represents the standard deviation. It is obvious that *I*_n_(*t*) recovered the most quickly in the non-gel aqueous solution. *I*_n_(*t*) of the non-compartmentalized bulky hydrogel rose more rapidly than that in the honeycomb microhydrogel network. This difference means that the molecules in the non-compartmentalized bulky hydrogel diffuse faster than those in the network. In other words, the molecular diffusion was hindered by the boundary of the honeycomb-shaped microhydrogel in the network, and we consider that most molecules will be trapped in the honeycomb-shaped microhydrogel. Nevertheless, slight molecular diffusion was observed from the outside of the honeycomb-shaped microhydrogel.

FRAP analysis yielded the observation that the diffusion of the molecules in the honeycomb microhydrogel network was slight and slow compared with the bulk hydrogels. If larger molecules are used, the molecules can be confined in the honeycomb-shaped microhydrogel. Our honeycomb microhydrogel network has the potential to provide a communication network based on the diffusion of small molecules synthesized by larger molecules confined in each honeycomb-shaped microhydrogel. Because each microhydrogel was densely packed and in close contact, the small molecules diffusing out of the microhydrogel would directly reach the neighboring microhydrogel and trigger synthesis reactions. Therefore, it would be possible to set up diffusion-based molecular communication, e.g., a network composed of the honeycomb-shaped microhydrogel that encapsulates cells itself and cell-free gene expression systems. In addition, in this network, the frequency of communication among the honeycomb-shaped microhydrogels could be controlled by controlling the packing degree (Figure 3), because the time of molecular diffusion is generally affected by the medium. In particular, when *R* is large (Figure 3a, *F*_G_ = 67× g), because there are many gaps between each microhydrogel, communication like quorum sensing [19] may be observed.

Cytotoxicity due to UV irradiation during the construction cannot be avoided when constructing a network including cells and biomolecules. However, it could be minimized by optimizing the experimental parameters, including the UV power and the concentration of the photosensitive polymer solution, or by using photosensitive polymers that can be solidified using a longer wavelength of light.

### 3.4. Fabrication of Triple-Layered Honeycomb Hydrogel Network

Figure 5a shows the construction of the triple-layered honeycomb microhydrogel network (left: single layer, middle: double layer, right: triple layer). The triple-layered structure was constructed as Figure 5a. The insets show fluorescent colors (upper) and simplified diagrams of the layered honeycomb microhydrogel network (lower). The boundaries and middle positions of layers are shown in Figure 5b. The upper panel shows CLMS images, and the black arrows in the lower illustrations show the observed position of the triple-layered structure. In the CLMS observation, a *zy*-plane of the cut triple-layered structure was observed. These images indicate that the boundaries were formed; thus, by the stacking of each layer, we successfully fabricated the triple-layered honeycomb microhydrogel network. On the other hand, in the second layer (red), the honeycomb packing did not form clearly, as compared with the other layers. This incompleteness may be attributed to the excessive deformation of the droplets, as discussed for the results in Figure 2b.

We consider that the first layer in the second left CLMS image in Figure 5b looked green because the aggregation of fluorescent beads was formed in the photosensitive polymer solution, resulting in a non-uniform fluorescent color. In addition, although the boundary of the layer appeared yellow (the second and fourth left CLMS images in Figure 5b), we speculate that this was because the color of the upper and lower layers overlapped. 

This triple-layered honeycomb microhydrogel network was sufficiently stable and did not break, even when it was manipulated using a tweezer (Figure 5c). This stable network will enlarge the range of experiments. For example, a dynamic and quick change of the environment surrounding the network could be easily achieved by transferring the network to another solution using a tweezer.

The triple-layered honeycomb microhydrogel network has the potential to reproduce the cell communication in multi-layered living tissue, such as blood vessels and skin. These tissues consist of layers formed by different cells, each of which has a different role. By stacking a variety of network layers, for example, layers including different functional molecules and cells, a multi-layered network with different functions can be constructed. Besides, the thickness of each layer can be controlled by the duration of centrifugation.

### 3.5. Fabrication of Network Structure Composed of Double-Faced Honeycomb Hydrogels

The fabrication of the network structure composed of double-faced honeycomb hydrogels is demonstrated in Figure 6a,b. It was confirmed that each of the honeycomb-shaped microhydrogels in the network was compartmentalized in two parts—the part containing red fluorescent microbeads and the part containing green fluorescent microbeads—as clearly shown in the high-magnification image (Figure 6b). Also, this demonstration suggests that a network composed of the multi-compartmentalized honeycomb-shaped microhydrogel with more than two compartments, may be realized using glass capillaries with more than two compartments [17,18].

The construction of the network composed of the multi-faced honeycomb-shaped microhydrogels is valuable. Such heterogeneous features are widely observed in biological systems. For example, epithelial cells exhibit spatial heterogeneities of channel proteins for efficient molecular transport. Honeycomb-shaped microhydrogels made heterogeneously from solutions with different concentrations could be used to realize a network with a heterogeneous diffusion profile. Furthermore, a combination of the multi-layered (Figure 5) and multi-faced (Figure 6) honeycomb microhydrogel network will be an ideal platform for mimicking the sophisticated functions of tissues.

## 4. Conclusions

We showed the controlled fabrication of the honeycomb microhydrogel network, and demonstrated the high stability and the suppression of molecular diffusion in the network. We constructed the honeycomb microhydrogel network utilizing the centrifugal force and the photopolymerization. In this method, moderately polymerized droplets were packed and deformed by *F*_G_, and were then completely polymerized, resulting in the formation of a honeycomb microhydrogel network with high stability. We also demonstrated the control of the network structure by changing *F*_G_. As we increased *F*_G_, the size of the individual honeycomb-shaped microhydrogels decreased, and the degree of packing increased. Additionally, it was confirmed that the molecular diffusion in the honeycomb microhydrogel network was inhibited by comparison with the bulk hydrogel. We also demonstrated the two types of fabrications of honeycomb microhydrogel networks with more complex structures. First, we fabricated the triple-layered microhoneycomb network, which had the distinct boundary of the differently fluorescent colored layers. A network composed of the honeycomb-shaped microhydrogel with two compartments was also fabricated. 

The honeycomb microhydrogel network demonstrated in this study was stable enough to be handled directly with a tweezer (Figure 5). Because a lipid bilayer behaves as a semipermeable membrane, the vesicles are sensitive to osmotic pressure [20,21]; thus, it is not easy to control the structures after the reaction in the vesicle network. In the droplet network, even the droplets were coated with lipids, and unexpected fusion sometimes occurred when an electric voltage was applied [9]. Contrary to these cases, because our network was composed of the hydrogel, the structure of each microhydrogel cannot be broken by osmotic pressures or applied voltage, etc. We believe that our highly-stable honeycomb-shaped microhydrogels network will help to elucidate functions of living multicellular systems orchestrated by cell communications.

## Figures and Tables

**Figure 1 life-08-00038-f001:**
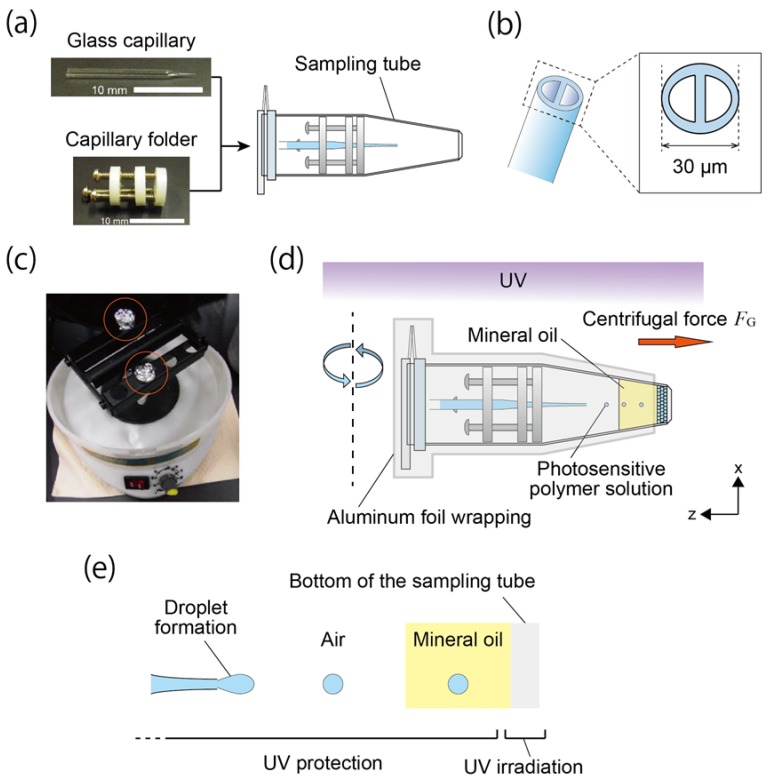
Centrifuge-based microfluidic device and fabrication scheme of the honeycomb microhydrogel network. (**a**) Setup of the centrifuge-based microfluidic device. The microfluidic device consisted of the theta-shaped glass capillary, the capillary holder and the sampling microtube. (**b**) Illustration of the theta-shaped glass capillary. A partition exists inside the capillary. The orifice diameter was tuned to 30 μm. (**c**) The device in the centrifuge. The orange circles indicate the device wrapped by aluminum foil. (**d**) Schematics of the fabrication of the honeycomb microhydrogel network. The centrifugal force *F*_G_ was applied to the assembled microfluidic device under UV irradiation from the top of the centrifuge. The honeycomb microhydrogel network was formed at the bottom of the sampling tube. (**e**) Formation and discharging of the droplet under the UV irradiation. Owing to *F*_G_, the photosensitive polymer solution in the glass capillary was dripped and discharged to the air phase. In the glass capillary, the polymerization of the photosensitive polymer solution was inhibited because of the partial protection of the UV irradiation.

**Figure 2 life-08-00038-f002:**
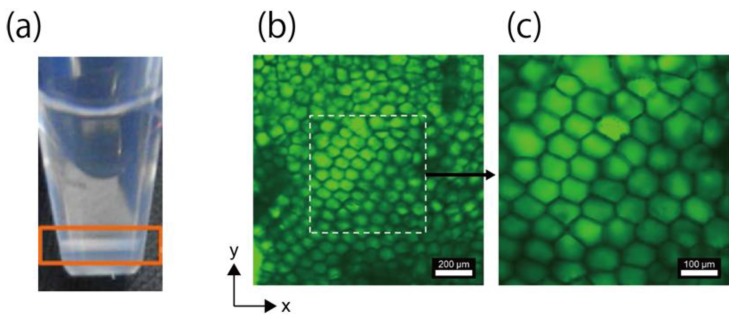
Observation of the honeycomb microhydrogel network. (**a**) Honeycomb microhydrogel network that accumulated at the bottom of the sampling tube (orange rectangle). (**b**) CLSM images of the honeycomb microhydrogel network. Many of the microhydrogels had the honeycomb shape and were densely packed. (**c**) Enlarged image of the dashed squared area in (**b**).

**Figure 3 life-08-00038-f003:**
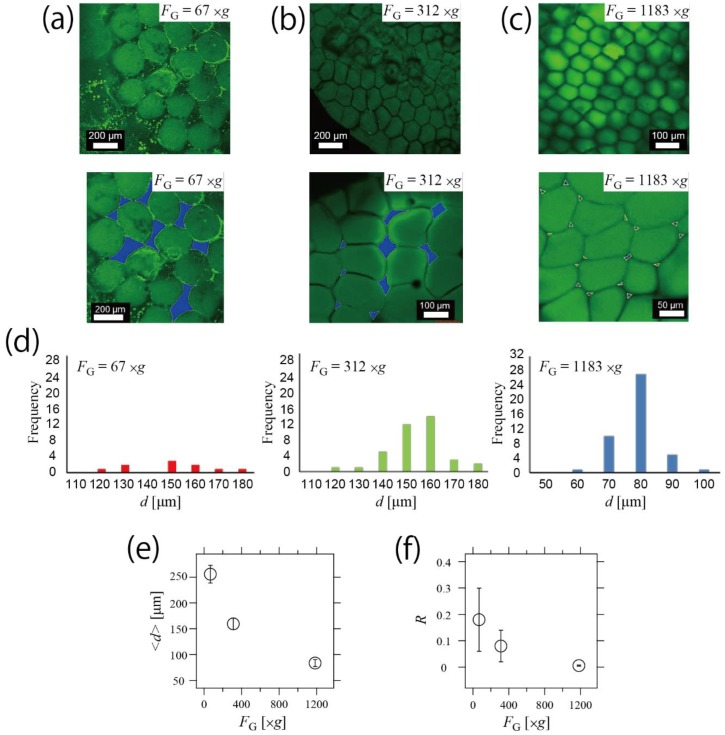
Control of the structures of the microhydrogel in the network by changing *F*_G_. (**a**–**c**) CLSM images of the honeycomb microhydrogel network fabricated under various values of *F*_G_. The applied *F*_G_ was 67× *g* (**a**), 312× *g* (**b**), and 1183× *g* (**c**). Lower images are magnified images. *S*_gap_ corresponds to the blue area in the lower images. It was speculated that the green puncta in the image of *F*_G_ = 67× *g* (**a**) were the emulsions of exuded fluorescein sodium salt solution from the microhydrogel. (**d**) Histograms of <*d*> of each *F*_G_. (**e**) Control of <*d*> by changing *F*_G_. <*d*> was decreased by increasing *F*_G_. (**f**) Control of *R* by changing *F*_G_. *R* was decreased by increasing *F*_G_.

**Figure 4 life-08-00038-f004:**
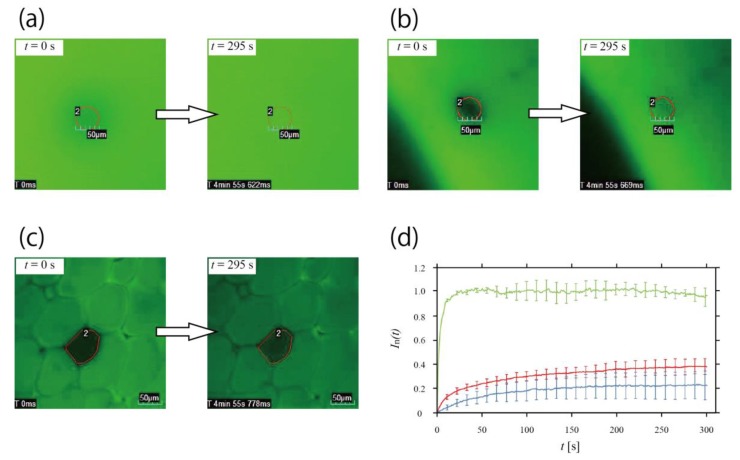
FRAP analysis in the honeycomb microhydrogel network. (**a**–**c**) Frames of the movie of the fluorescence recovery. FRAP analysis was performed in (**a**) the non-gel aqueous solution, (**b**) the non-compartmentalized bulky hydrogel, and (**c**) the honeycomb microhydrogel network fabricated under *F*_G_ = 1183× *g*, and the fluorescence recovering was observed for 5 min. The area enclosed by the red line is the photobleaching area. The numbers near each area were for analysis. (**d**) Time course of *I*_n_(*t*). *I*_n_(*t*) of the honeycomb microhydrogel network (blue curve) had a longer recovery time than that of the non-gel aqueous solution (green curve) and the non-compartmentalized bulky hydrogel (red curve).

**Figure 5 life-08-00038-f005:**
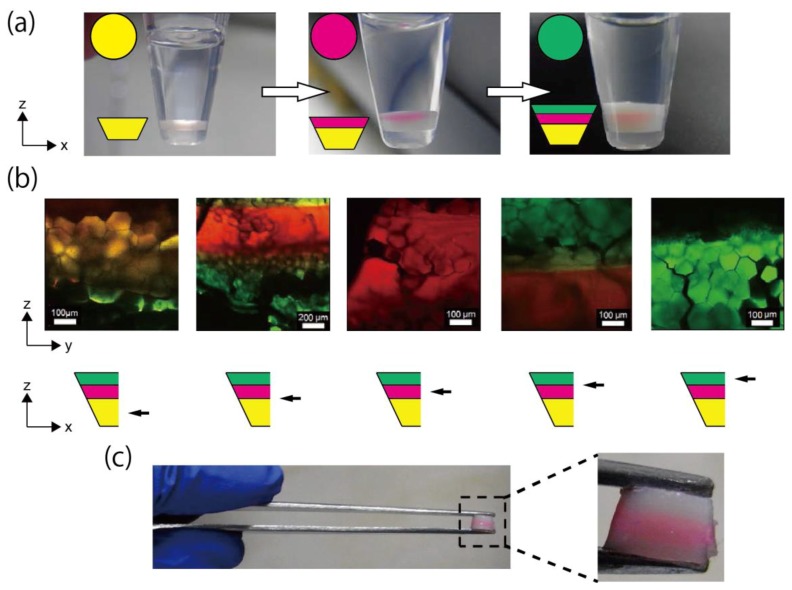
Fabrication and observation of the triple-layered honeycomb microhydrogel network. (**a**) Fabrication process of the triple-layered honeycomb microhydrogel network. Initially, the first layer was formed at the bottom of the sampling tube (left). Then, the second layer was formed on the first layer (middle). Finally, the third layer was formed on the second layer (right). All layers were fabricated under *F*_G_ = 1183× *g*. Upper insets: fluorescence coloring of the layers. Lower insets: simplified diagram of the layered honeycomb microhydrogel network. (**b**) CLMS images of the *zy*-plane of the cut triple-layered honeycomb microhydrogel network and illustrations of the observed position. The images in the upper panel are CLMS images of the *zy*-plane of the triple-layered structure. The observed position in the triple-layered structure is shown by the black arrows in the illustrations of the lower panel. (**c**) Direct handling of the triple-layered honeycomb microhydrogel network. The network was easily handled by a tweezer.

**Figure 6 life-08-00038-f006:**
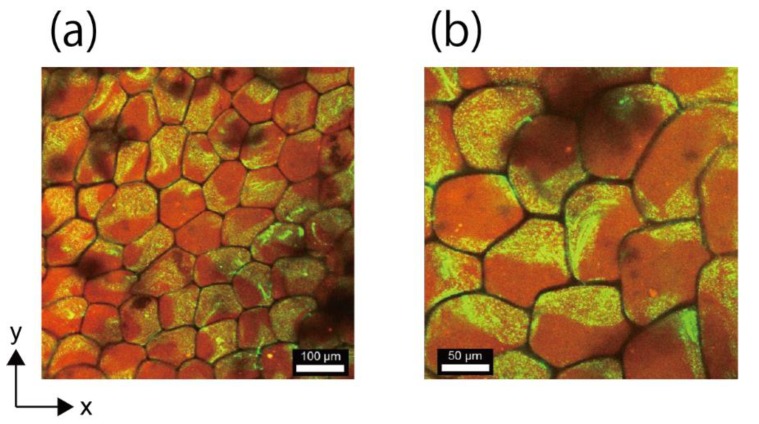
Observation of the network structure composed of the double-faced honeycomb-shaped microhydrogels. (**a**,**b**) CLSM images of the network. Each of the honeycomb-shaped microhydrogels in the network was compartmentalized in two parts. Applied *F*_G_ was 1183× *g*.

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
