# Peer review of "Controlled Construction of Stable Network Structure Composed of Honeycomb-Shaped Microhydrogels"

_life, 2018, doi:10.3390/life8040038_

Round 1
Reviewer 1 Report
The manuscript by Hayakawa et al. seeks to develop a method to construct a stable network of honeycomb-shaped microhydrogels using a centrifugal-based approach with UV curable hydrogel. The main target of the author’s interest is to use this as a model of cell communication in multicellular systems and the authors argue that current systems using droplets and vesicle networks are unstable. The basic principle of microhydrogel generation is based on previously published articles from the same group and others (Refs 15 and 16) with the modification that replaces alginate gel with UV curable polymer. From a methodology standpoint, the novelty lies in the combination of photosensitive polymer with centrifugal-generated droplets. There are some characterization and demonstration using the system. The writing is overall clear, but there can be significant improvements in several areas before this can be published.
Major comment:
- My biggest issue with this work is the quality of the images. The honeycomb microgel structure is most clearly defined in the inset of Figure 2c, and in parts of Figure 3a. What are the green puncta to near the bottom left of the image in 3a? I did not see any discussion on the homogeneity of droplet size and is it possible that the structure doesn’t look so good because some droplets are out of the plane? It is certainly not unreasonable given the gels are cured and packed at the same time that the pattern will not be perfect. All the images in Fig 5b lost the honeycomb structure, is it due to the fact that this is the z-y plane? The second set of images in Fig 5b should be the yellow-red interface, but the bottom showed green instead (the top looked more yellow). Is that due to a mistake? I would recommend the authors to replace with better images (if there are available) and comment on these imperfection of honeycomb structures. In general, there is a lack of discussion on the results presented in this work.
- One of the main arguments that the authors made about this system is that this microhydrogel network is more stable, compared to previous droplet and vesicle network. I don’t see any distinguished features of this gel network, which are also made of droplets, that are more stable than what others have done. Figure 5 shows that the gels can be handled by a tweezer and it may be good to mention this earlier as I was unsure how the gel was imaged in earlier Figures (done in the tube or removed from the tube). Given stability is a challenge the authors discussed as drawbacks of current systems, the authors should discuss this aspect more.
- For the FRAP analysis of microhydrogel network, I think it will be very difficult to make a convincing argument that the bulk hydrogel is a proper control to compare with. Besides the 1-minute UV illumination, are the hydrogels identical? If they are polymerized differently, that can influence diffusion of dye. The authors can perform bleaching expts on more bulk hydrogels with different photopolymerization time. The example did not show the bulk hydrogel images and the writing did not make it clear enough what the authors want to say about this experiment, due to the lack of discussion. There was no information on how many measurements were made to generate the graph in Fig 4d and what the error bars were. Are there statistical differences between the bulk hydrogel and microhydrogel? I would assume the microhydrogel was also bleached by a 50 µm spot, but the outline of the microhydrogel would be misleading this point. Relating this experiment and other demonstrations that the authors have provided, it would be much more informative if they be can tied to cell communication in multicellular systems. E.g. What is the rationale for making multi-compartmentalized honeycomb hydrogels? Personally, I do not see how this experimental platform would have much utility in studying multicellular communication. Perhaps, if cells were co-encapsulated in the droplet and the different layers of two-faced droplet of cells/diffusible compounds are used, this could be useful. This should be better discussed to provide a better context for the experiments.
Minor comments:
- It was not at all clear to me why theta pipettes were used in this work. From what I can understanding, the same solutions were loaded in both channels for most of the work, except for the two-faced microgels, correct? It may be helpful to make it clear at the beginning to justify why theta pipette was used.
-More details should be included on how Sgapand Sgelare measured. Is this by intensity thresholding? I noticed in some images that outlines of the droplet were marked. If this is done by software, it should also be described. Why say ‘assumed’ diameter of honeycomb? Maybe consider a different word. Can the authors comment on the apparent size distribution of this network? It would be easy to plot a histogram of all the Sgel.
- As mentioned earlier, it should be noted early on that these microhydrogel can be removed from the sampling tube by a tweezer and be imaged on a microscope
- How UV light is applied should be better described, i.e. where is the lamp placed (what is the power?), is it over the entire setup or just over a specific spot as the sample spins.
- Sentences 54 – 57 are not so clear. It appears that the authors work also don’t address the limitation mentioned. But maybe I don’t fully understand what the authors are trying to say here.
- Line 102: it may be helpful to point out right away that each layer has multiple layers. When I first read this, I had the impression that there are three layers, which I know would be difficult to control here.
- Line 108: is there any reason why the red fluorescent bead is 10% of the green fluorescent bead?
- Line 115: Rational for adding sodium alginate is strange. What is so special about this to prevent solutions from mixing with each other? Sodium alginate is used to make hydrogel as the authors have demonstrated in the past.
-For Fig 3a, it'd be best to remove the numbers stamped on the image, it can be mistaken as something real given they are too small to see. I would say the same is true for ‘2’ (what does 2 mean here anyways?) and the 50 µm reticule from Figure 4 (using a scale bar is better).
- Figure 6 should describe what are the different probes shown in this figure. The centrifugal speed should also be in the figure caption for Figures 4, 5 and 6.
Writing: Writing is generally clear, but there were a few awkward phrases. I will point out a few errors below.
- Abstract first sentence and line 47: ‘has been developing’ is awkward and needs to be revised.
- Line 48 and line 65: replace clarifying with something else.
- Line 51: remove consisting.
- Line 59: revise well control
- Line 61: no ‘a’ before molecular diffusion
- Line 87: fluorescence instead of fluorescent
- Line 104: ‘was renewed’ is an awkward description here, consider revising this.
- Line 166: no ‘the’ before molecular diffusion
Author Response
Please find attached pdf file.

Reviewer 2 Report
Hayakawa et al use a centrifugal microfluidic device with photopolymerizable polymers to generate a hydrogel network that they claim is useful for the construction of multicellular, tissue-like systems. The microfluidic device is quite interesting but has bee previously published. The authors are able to control the shape and properties of the hydrogel by centrifugal force.
In my opinion, the work is fine. However, it wasn't always clear from the writing what was new and what was done before. Also, the work does not address the compatibility with any of the material that would be needed to make models of multicellular systems. Would transcription and translation systems work in these hydrogels. Would they survive the process of generating these hydrogels? Would even a protein enzyme retain activity? This lack of a practical connection to stated goal of making models of tissues was a bit frustrating.
The authors rightly reference Bayley for his 3D printed droplet, tissue-like systems. However, there are other studies that explored communication between droplets, such as described in papers from Schwarz-Schilling M et al and Lentini R et al.
Even if it's not directly related to what's presented, I can't help but wonder what would happen if the polymerization is carried out during the phase in which the droplet is in the air, before hitting the mineral oil. This is not a criticism, just a comment.
I do think that this work is useful to the field. If the described device is easy to use, then I could see others that don't have expertise in microfluidics wanting to use the described system.
Round 2
Reviewer 1 Report
The authors have address my comments and implemented relevant changes in the manuscript.